# Challenges in Drug Surveillance: Strengthening the Analysis of New Psychoactive Substances by Harmonizing Drug Checking Services in Proficiency Testing

**DOI:** 10.3390/ijerph20054628

**Published:** 2023-03-06

**Authors:** Margot Balcaen, Mireia Ventura, Cristina Gil, Anton Luf, Daniel Martins, Mar Cunha, Karsten Tögel-Lins, Danny Wolf, Peter Blanckaert, Eric Deconinck

**Affiliations:** 1Unit Illicit Drugs, Lifestyle and Chronic Diseases, Scientific Direction Epidemiology, Sciensano, 1050 Brussels, Belgium; 2Energy Control, Associació Benestar i Desenvolupament, 08041 Barcelona, Spain; 3Clinical Department of Laboratory Medicine, Medical University of Vienna, Waehringer Guertel 18–20, 1090 Vienna, Austria; 4Kosmicare, 1170-283 Lisbon, Portugal; 5Legal-high-Inhaltsstoffe, 60439 Frankfurt, Germany; 6Service Medicines and Health Products, Scientific Direction Physical and Chemical Health Risks, Sciensano, J. Wytsmanstraat 14, 1050 Brussels, Belgium

**Keywords:** new psychoactive substances, drug checking, toxicology, harm reduction, substance use, laboratory testing

## Abstract

Background: Drug checking is a proven harm reduction strategy and provides real-time information on the market of new psychoactive substances (NPS). It combines chemical analysis of samples with direct engagement with people who use drugs (PWUD), giving the ability to increase preparedness and responsiveness towards NPS. Next to that, it supports rapid identification of potential unwitting consumption. However, NPS cause a toxicological battle for the researchers, as factors such as the unpredictability and quick shift of the market complicate the detection. Methods: To evaluate challenges posed towards drug checking services, proficiency testing was set up to evaluate existing analytical techniques and investigate the capability to correctly identify circulating NPS. Twenty blind substances, covering the most common categories of substances, were analyzed according to the existing protocols of the existing drug checking services, including several analytical methods such as gas chromatography–mass spectrometry (GC-MS) and liquid chromatography with diode array detector (LC-DAD). Results: The proficiency test scores range from 80 to 97.5% accuracy. The most common issues and errors are mainly unidentified compounds, presumably due to no up-to-date libraries, and/ or confusion between structural isomers, such as 3- and 4-chloroethcathinone, or structural analogs, such as MIPLA (N-methyl-N-isopropyl lysergamide) and LSD (D-lysergic acid diethylamide). Conclusions: The participating drug checking services have access to adequate analytical tools to provide feedback to drug users and provide up-to-date information on NPS.

## 1. Introduction

The emergence of new psychoactive substances (NPS), ‘legal’ alternatives to internationally controlled drugs, also known as ‘designer drugs’ or ‘legal highs’, repeatedly challenges drug surveillance and control. Each year, about 50 new substances are encountered on the European market for the first time. For more than 25 years, the European Monitoring Centre for Drugs and Drug Addiction (EMCDDA) has been sharing information on NPS within its network, including the European (EU) Early Warning System (EWS) [1,2]. As this market has proven to be highly unpredictable and contains a wide variety of substances, there is an overriding need to share up-to-date information on designer drugs. For many countries, it is becoming a priority to focus on proactive systems for collecting information on the identity, sale and associated consumption of NPS [3]. Strengthening the limited data will seek answers to questions such as: What substances slip through the cracks? Which NPS pose a public health risk? What is the prevalence of legal highs [4]?

The innovation and responsiveness of the NPS market are forcing consideration of different types of data sources to estimate the occurrence of NPS more accurately [3,5]. Traditional approaches, such as self-reported use surveys, are being replaced by new methods of gathering information, as surveys involve a potential delay between data collection and reporting. Moreover, they do not always cover new substances but mainly established drugs [6,7]. Since the NPS market is shifting rapidly, this may mean that the findings do not reflect the current availability of NPS. When investigating alternatives, drug checking services (DCS) appear to generate real-time information on NPS quickly, as substances directly derived from people who use drugs (PWUD) are analyzed within hours to days of submission [8,9,10]. Not to be neglected are other complementary methods, such as wastewater monitoring, analyses of drug-specific excretion products from human sewerage, coronial data, and analysis of drug toxicity data at autopsy [11,12,13]. Although these monitoring techniques are valuable, they cannot guarantee the same rate of results as DCS [5]. An additional advantage of DCS is the rapid identification of potential unwitting consumption, where PWUD are unaware of the true identity of their drugs. This can occur, for instance, if the assumed identity under which the sample is sold online does not match the true identity. Another example is drugs adulterated with potentially dangerous or lethal substances, e.g., Xanax pills adulterated with etizolam [9,14].

As drug checking services combine chemical analysis of samples with direct engagement with PWUD, this has proven to be a well-working harm reduction strategy [15,16]. The latter has recently become more prominent as regulatory measures, which seek to control drug use and supply, have little effect on public health issues arising as a consequence of NPS use [3]. Therefore, priority has been given to evidence-based harm-minimization approaches, of which DCS are a prime example. Harm reduction includes interventions and programs, such as the provision of opioid substitution treatment, needle and syringe programs to reduce the prevalence of infectious diseases, and tools such as urine fentanyl test strips [17,18,19].

Information produced by DCS can be used at different levels, starting with, but not limited to, issuing alerts and risk communication at the national level. Trends often transcend national borders; therefore, in order to ameliorate preparedness and responses towards emerging NPS, available information is shared within several established networks, such as the EU EWS of the EMCDDA and the Trans European Drug Information (TEDI) network [1]. TEDI unites 20 services from 13 different European Countries (Austria, Belgium, France, Germany, Italy, Luxembourg, Portugal, Slovenia, Spain, Switzerland, the Netherlands, Finland, and the United Kingdom), combining the data generated by all participating DCS, which allows creating a global picture of the drug market in Europe. This network has recently shown the value of comparative market monitoring at the European level, such as occurred concerning the increase in MDMA tablet potency observed between 2012 and 2021. Drug checking services showed that this was due to an increase in tablet weight and not an elevation in the ratio of active ingredient to filler. That this result was observed across all participating TEDI DCS, despite the diversity of technologies used, reinforces its robustness and shows its value within harm reduction [20,21]. The European project SCANNER harmonizes four of these drug checking services, this being Energy Control (Spain), Kosmicare (Portugal), Check-it (Austria) and Legal-high inhaltstoffe (Germany). The project, with as main objective of understanding the dynamics and consequences of NPS in a rapidly changing (online) market, also contains a work package evaluating NPS sold online and improving analytical quality within drug checking services.

The need to improve chemical analysis has its origin in the significant challenges NPS poses for clinicians, not only those working at a drug checking service, but also for researchers, forensic toxicologists, and others in general. Keeping up with the rise of this ever-growing group of substances has proven to be a continuous toxicological battle for researchers [16,22]. In addition, the detection of NPS is complicated by the chemical diversity, market unpredictability and the swift emergence and potentially yet faster disappearance of these drugs [23]. To evaluate these challenges, selected and purchased samples of NPS are analyzed in a proficiency testing (PT), also known as a ring test. 

The proficiency testing serves as an evaluation of existing analytical techniques, such as mass spectrometry, applied within the DCS through inter-laboratory comparison of applied techniques and corresponding results. It answers the question of whether these services are capable of analyzing NPS with the available resources. By purchasing and subsequently analyzing NPS, the test will also establish the extent to which the identity of NPS available online matches the presumed identity as advertised by the vendor. This paper aims to provide an overview of the analytical challenges faced by DCS, followed by possible solutions and recommendations. Together with guidance on which methods are essential when analyzing NPS, it can serve as the basis for other countries or governments wishing to implement DCS.

## 2. Materials and Methods

### 2.1. Criteria for Inclusion

The substances included in the proficiency testing were selected by the following criteria:Origin: Substances purchased from online (clear net) markets (or) received as ‘unknown’.Chemical and pharmacological features:○All compounds belonging to the most prevalent categories of NPS reported to EMCDDA, i.e., cannabinoids, cathinones, opioids, tryptamines, phenethylamines and benzodiazepines [2].○A heterogeneous group of substances covering a broad spectrum of classes. See Figure 1, which shows the share of each class of NPS in the proficiency testing based on the EMCDDA classification. (See Figure 1).Analytical requirements: taking into account the recommended scope for laboratories testing on NPS (quarterly updates) as the basis for the purchase of the NPS [24,25,26,27].

### 2.2. Proficiency Testing

The design of the proficiency test was based on the aim to evaluate the capacity of DCS to analyze NPS. Because of this, there was no fixed protocol with a procedure to follow. Instead, all participating services could apply their own protocol, as the resources DCS have at their disposal differ. It was mandatory to handle a sample, as it is treated in the daily operation of the DCS itself. The coordinator of the proficiency testing performed the analysis prior to shipment to have certainty on the identity of the samples. The chosen substances were then sent as unknowns, with a unique code in the sign of reporting the results. The four participating laboratories (Energy Control, Kosmicare, Check-it and legal-high inhaltstoffe) received three shipments of 20 samples: batch 1 containing two samples, batch 2 including six samples and batch 3, consisting of 12 samples. The deadline for these batches was set depending on the batch size and could go up to two months. After the submission of the results by the participants, a review meeting was convened by the coordinator to thoroughly discuss the results and go over associated challenges and possible recommendations for future analysis.

Depending on the available budget, the techniques used within DCS range from screening to additional confirmatory analysis. Table 1 provides an overview of the methods used in proficiency testing. The techniques in Table 1 without a special sign are included in the standard protocol of the laboratory. When necessary, the analytical methods containing a * or ° are applied, e.g., if, after gas chromatography–mass spectrometry (GC-MS), the score match to the library is not sufficient. Table 2 contains all results of the proficiency testing. It is pointed out for which samples standard protocol was sufficient and when additional techniques were added to the analysis. 

## 3. Results

The results of the three batches of samples included in the proficiency testing can be found in Table 2. Wrong results are pointed out in bold in Table 2. 

### 3.1. Batch 1

Sample 1A, beta-hydroxy 2c-b (BOH-2C-B), was identified as such by all four laboratories. Sample 1B, purchased as Cumyl-Pegaclone (or SGT-151), was identified by three laboratories as Furanyl UF-17. For service L4, this sample resulted in U-47109.

### 3.2. Batch 2

For sample 2A, purchased as SGT-263, 5-chloro-AKB48 (5C-AKB48) was identified as the main component. Three of the participating laboratories also detected the presence of 4-cyano-cumyl-botanica (CUMYL-4CN-BINACA). Sample 2B resulted in 4-fluoro-MDMB-BUTINACA (4F-MDMB-BINACA), corresponding to the label of the sample. Sample 2C, purchased as MIPLA, was identified as such by three laboratories. L2 reported it as the structural analogue LSD. Sample 2D was identified as Brorphine by all participating services. Sample 2E, which contained Butonitazene, was detected as such by three of four laboratories. Isotonitazene, an analogue with a structural difference on the O-alkyl chain, was reported by L2. The last sample of this batch, 2F, was identified as etonitazepyne by all laboratories but one. The latter was unable to identify the content.

### 3.3. Batch 3

All laboratories identified MDMB-4-en-PINACA as the main active ingredient of sample 3A, corresponding to the label 5-Cl-ADB-A. Laboratory L1, L2 and L3 also identified the additional presence of 4F-MDMB-BINACA. In 3B, Flualprazolam was detected by all four services, consistent with the labelling of the sample. The unlabeled sample 3C was matched with Isopropylphenidate by all laboratories. Sample 3D contained 4-chloro-alpha-PVP (4-Cl-α-PVP), 3 or 4-chloroethcathinone (3 or 4-CEC), 3 or 4-chloromethcathinone (4-CMC) and 4-chlorpentedrone according to all services. For sample 3E, two possibilities were reported, namely U-51754 or U-48800. Unlike the other services, for sample 3F, L4 reported x-chloroethcathinone since 4-CEC could not be confirmed. The unlabeled sample, 3G, was identified by all laboratories as dibutylone (or Bk-DMBDB). 3H, with ‘BC66’ on the bag, was identified as 4-fluoro-MDMB-BUTICA (4F-MDMB-BICA). Everyone identified sample 3I as 3F-PCP, in accordance with sample labelling. 3J, purchased as Fluonitazene, was identified as such by all services. Samples 3K and 3L were identified by three of the four participating laboratories as Hydroxetamine (HXE) and Deoxymethoxetamine (DMXE). Laboratory L3 could not identify these compounds.

## 4. Discussion

The general image of drug checking services is that they work with test kits (often based on colorimetry), test strips, thin-layer chromatography and possibly spectroscopic methods to analyze samples brought in by PWUD. In this case, DCS would be limited to the analysis of classical drugs and some targeted adulterants or contaminants [28,29]. Although many drug checking services started this way, the participating services have access to adequate analytical resources to identify new psychotropic substances to a certain level. This statement is supported by the proficiency test scores, ranging from 73.8% to 97.5% accuracy. Most services rely on gas chromatography-mass spectrometry (GC-MS) and/ or liquid chromatography-mass spectrometry (LC-MS) for analysis. The most common issues and errors that occur are unidentified/incorrect compounds (57.1% of all errors) and confusion between structural isomers and/or structural analogs (42.9%). Important to note is that the highlighted issues and recommendations listed below are based on the limited proficiency testing of 20 samples. Regular repetition of the proficiency testing is advised to bring all analytical difficulties to light and further ameliorate the analysis of NPS.

Before going into depth on the errors occurring within the PT, it should be noted that the presence of NPS is generally a red flag and requires quick response and corresponding feedback to the person presenting the sample for analysis. The appearance of positional isomers x-chloroethcathinone, e.g., 3-CEC vs. 4-CEC, or other structurally related substances, is considered severe. Adaptations to existing techniques should be considered to improve general analysis. However, if there is no readily available method for the distinction between two similar components, PWUD should be informed appropriately about the possible content of the substance. Testing pre-consumption has been proven to lead to safer drug user behavior, including reducing doses and not using alone [30].

Some laboratories reported misidentifications or could not detect certain substances. This could be due to several factors; the most plausible is the absence of the molecules in spectral libraries, e.g., etonitazepyne (sample 2F) first appeared on the market when proficiency testing was ongoing, whereas isotonitazene was already well-known [31,32]. A regular update of libraries could partially solve this problem. However, it should be noted that the time it takes for an NPS to be present in the libraries is dramatically larger than to be sold in the streets. Another explanation could be the interference of other substances in the sample. This again highlights the importance of proficiency testing in this context.

Concerning the distinction between structural isomers (e.g., sample 2C, 3D and 3E), who have identical molar masses and a quasi-identical fragmentation pattern, the participating DCS employed several approaches. For sample 2C, MIPLA versus LSD (see Figure 2), the distinction was made on the basis of retention times. These were already included in the libraries for one laboratory from prior analysis with reference standards [33]. It is important to highlight that purchasing analytical standards is often quite expensive, and not all services have the resources to do this. Next to that, quick market changes do not allow DCS to wait on standards, as administrative procedures and shipment lead to long waiting times [34]. Adding retention data to MS libraries is not self-evident, as this requires regular adjustments of the existing method in the sign of newly added components. Adaptations include altering oven temperature and program, injection temperature or flow rate of carrier gas, which does not favour a fast and efficient workflow.

An alternative for distinguishing cathinones, Liquid Chromatography-Diode Array Detection (LC-DAD), was applied by one of the participating services. Based on the discrepancies between UV spectra, compared with an existing in-house database, it was possible to differentiate between the positional isomers 3- and 4-chloroethcathinone (sample 3D, see Figure 2). Whilst being a good solution towards the analysis of isomers, it is not easily feasible, as it adds a supplementary technique and related cost to the analysis routine. Often, screening methods are being developed to detect a broad spectrum of compounds, e.g., 64 different NPS [23]. Although these methods add value towards the detection of NPS, they require certain instruments and equipment and are often quickly obsolete due to the swift dynamics of NPS.

In the case of sample 3E, it is, for some services, impossible to differentiate between structural analogues, as the two molecules, U-48800 and U-51754 (see Figure 2), differ only in the position of a chloride atom (regioisomers) [35,36]. In this case, services should have access to nuclear magnetic resonance (NMR) analysis, e.g., through collaboration with a university laboratory or governmental laboratory [37,38,39]. However, given the significant cost and high turnaround time, the implementation of NMR is not always feasible [40].

The proficiency testing also demonstrated the occurrence of plausible unintended consumption [5]. In the case of sample 1B, a U-type opioid, Furanyl Uf-17, is sold as the gamma-carboline-1-one core containing synthetic cannabinoid cumyl-pegaclone (SGT-151) [41]. Furanyl UF-17 belongs to the U-compounds, which are non-fentanyl, opioid-related substances, that function as potent and efficacious μ-opioid receptor (MOR) agonists, often more potent than heroin or morphine [42,43]. Although SGT-151 also appears to be highly potent, indicated by the agonist activity at cannabinoid receptor 1, its mechanism and effects are not comparable with Furanyl UF-17. Similar is sample 2A, whose presumed identity was SGT-263, but in which the presence of 5C-AKB48 and CUMYL-4CN-BINACA was detected. Although all three substances belong to the class of synthetic cannabinoids, the latter is a controlled drug listed in the 1971 Convention on Psychotropic substances [44]. In this case, a ‘legal’ substance is sold containing a substance under international control. Finally, proficiency testing revealed the adulteration of a sample. Sample 3A, sold online as 5-Cl-ADB-A, contains not only MDMB-4-en-PINACA but 4F-MDMB-BINACA as well. Since three of 20 samples involved the adulteration of samples or samples where the presumed identity does not match the real content, we can state that unwitting consumption can be common and regular testing can provide valuable information on this. 

## 5. Conclusions

Overall, it can be concluded that the participating drug checking services have all the necessary analytical tools to characterize psychotropic samples containing NPS, so that an initial risk assessment can be carried out and feedback can be provided to PWUD. All this is primarily for harm reduction purposes but also useful for drug policy in broader terms. As mentioned, based on the results of this limited proficiency testing, a few analytical recommendations can be made when implementing drug checking services. The basis to adequately analyze NPS is to combine different analytical laboratory techniques at own premises or through collaboration with other laboratories or universities and the presence of up-to-date open-source MS libraries. When going more in-depth on the possible adaptations/solutions towards the occurring problems, the following ideas can be put forward:(i)Each laboratory can extend its GC and/or LC library with retention data from characterized samples or reference standards to distinguish between stereo-isomers. Based on this, the screening methods should ‘evolve’ to separate structural isomers. As an alternative, laboratories can use reference standards or confirmed samples to create a UV library, to identify isomers with classic LC-DAD analysis or, taking it even further, invest in an LC-MS where a DAD is put in series before the MS detector. (ii)Since most participants make use of GC-MS as the first choice technique for the screening of NPS, a uniform standardized method could be proposed allowing the exchange of retention data for certain molecules. An ‘internal’ standard molecule could then be chosen in order to inject with all samples, allowing to work with relative retention times and correct for any retention shifts due to small technical differences (instrument, column age) and environmental influences.

The importance of building a network and collaborating has once again proven to be a key tool in the effort to close the growing gap in knowledge about emerging NPS. Furthermore, repetition of proficiency testing, combined with research and purchases on Clear and Deep web markets, will support laboratories in further tailoring their analysis, give the ability to continuously monitor the market and remain ever vigilant towards the ever-increasing predominance of NPS. 

## Figures and Tables

**Figure 1 ijerph-20-04628-f001:**
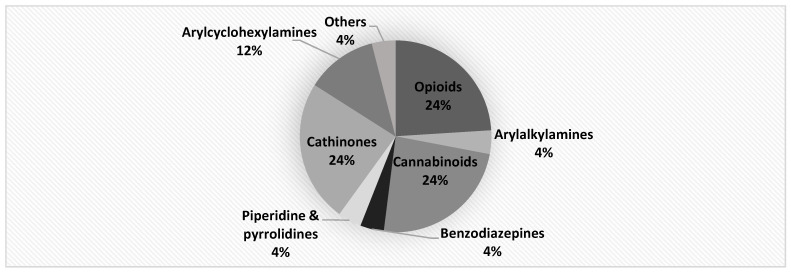
Share of classes of new psychoactive substances represented in the PT.

**Figure 2 ijerph-20-04628-f002:**
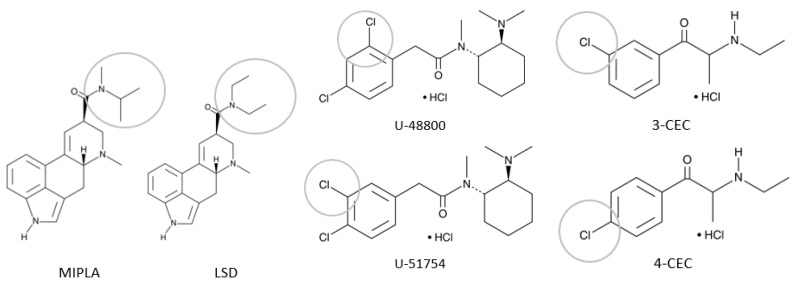
Structural analogs/isomers: MIPLA vs. LSD, U-48800 vs. U-51754 and 3-CEC vs. 4-CEC.

**Table 1 ijerph-20-04628-t001:** Methods used by the participating laboratories.

Laboratory	Technique
L1	GC-MS LC-MS *
L2	GC-MS FTIR *
L3	LC-MS MALDI-HR-MS FTIR * LC-DAD °
L4	GC-MS

GC-MS = Gas chromatography–mass spectrometry. LC-MS = Liquid chromatography-mass spectrometry. MALDI-HR-MS = Matrix-Assisted Laser Desorption/Ionization High-Resolution Mass Spectrometry. FTIR = Fourier-transform infrared spectroscopy. LC-DAD = Liquid chromatography with diode-array-detector.

**Table 2 ijerph-20-04628-t002:** Results of the three batches included in the proficiency testing.

	Bought/Received as:	L1	L2	L3	L4
Batch 1					
1A	BOH-2C-B	BOH-2C-B	BOH-2C-B	BOH-2C-B *	BOH-2C-B
1B	**CUMYL-PeGACLONE** **(or SGT-151)**	Furanyl UF-17	Furanyl UF-17	Furanyl UF-17 *	**U-47109**
Batch 2					
2A	**CUMYL-5F-P7AICA** **(or SGT-263)**	5C-AKB48 *	5C-AKB48 **CUMYL-4CN-BINACA**	5C-AKB48 *°	5C-AKB48 **CUMYL-4CN-BINACA**
2B	4F-MDMB-BINACA (or 4F-ADB)	4F-MDMB-BINACA *	4F-MDMB-BINACA	4F-MDMB-BINACA *°	4F-MDMB-BINACA
2C	-MIPLA	MIPLA *	**LSD** *	MIPLA *°	MIPLA
2D	Brorphine	Brorphine	Brorphine	Brorphine *°	Brorphine
2E	Butonitazene	Butonitazene	**Isotonitazene**	Butonitazene *	Butonitazene
2F	Etonitazepyne	Etonitazepyne *	**Unknown**	Etonitazepyne *	Etonitazepyne
Batch 3					
3A	5-Cl-ADB-A (or MDMB-4-en-PINACA)	MDMB-4-en-PINACA * **4F-MDMB-BINACA** *	MDMB-4-en-PINACA * **4F-MDMB-BINACA** *	MDMB-4-en-PINACA *° **4F-MDMB-BINACA** *°	MDMB-4-en-PINACA
3B	Flualprazolam	Flualprazolam	Flualprazolam *	Flualprazolam *°	Flualprazolam
3C	Isopropylphenidate	Isopropylphenidate	Isopropylphenidate *	Isopropylphenidate *°	Isopropylphenidate
3D	**Unknown**	4-CEC 4-Cl-α-PVP 4-chloropentedrone **4-CMC**	4-CEC * 4-Cl-α-PVP * 4-chloropentedrone *	4-CEC ° 4-Cl-α-PVP ° 4-chloropentedrone ° **4-CMC** °	**x-CEC** 4-Cl-α-PVP 4-chloropentedrone **x-CMC**
3E	**Unknown**	U-48800 **or U-51754** *	**U-48800** *	**U-48800** *°	U-48800 **or U-51754**
3F	**Unknown**	4-CEC	4-CEC *	4-CEC *°	**x-CEC**
3G	**Unknown**	Bk-DMBDB	Bk-DMBDB *	Bk-DMBDB *°	Bk-DMBDB
3H	BC-66	4F-MDMB-BICA	4F-MDMB-BICA *	4F-MDMB-BICA *°	4F-MDMB-BICA
3I	3F-PCP	3F-PCP	3F-PCP *	3F-PCP *°	3F-PCP
3J	Fluonitazene	Fluonitazene *	Fluonitazene *	Fluonitazene *°	Fluonitazene
3K	Hydroxetamine	Hydroxetamine *	**Unknown** *	Hydroxetamine *°	Hydroxetamine
3L	Deoxymethoxetamine	Deoxymethoxetamine *	**Unknown** *	Deoxymethoxetamine *°	Deoxymethoxetamine

Table 1 explains all applied methods carefully. If no special signs are added to a result, standard protocol was sufficient for analysis. Additional techniques used during analysis are pointed out using *, *° and/ or °. See Table 1 for the additional techniques. Wrong results are pointed out in bold.

## Data Availability

The data presented in this study are available on request from the corresponding author. The data are not publicly available due to the fact that the project has not been published by the European Commission yet. When available, the results will be published on the website of the project: home|NPS scanner (scannernps.eu).

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
