# Peer review of "Challenges in Drug Surveillance: Strengthening the Analysis of New Psychoactive Substances by Harmonizing Drug Checking Services in Proficiency Testing"

_ijerph, 2023, doi:10.3390/ijerph20054628_

Round 1

Reviewer 1 Report

The authors have described the challenges in drug surveillance. They pointed out the strengths and improvements needed for drug-checking services. It is an interesting work, but there are some comments:

i) In the Abstract, please describe the abbreviation for <<MIPLA>> and <<LSD>>;

ii) In the Discussion, the authors have mentioned that the most common issues and errors were unidentified compounds and confusion between structural isomers and/or structural analogs. Is it possible to present these data as numbers (%)?

iii) In the Conclusion, please correct the typo "standard";

iv) Why is not established a fixed protocol for these analyses?

Author Response

i) In the Abstract, please describe the abbreviation for <<MIPLA>> and <<LSD>>;

The abbreviations are now described in the abstract. 

ii) In the Discussion, the authors have mentioned that the most common issues and errors were unidentified compounds and confusion between structural isomers and/or structural analogs. Is it possible to present these data as numbers (%)?

Thank you for the suggestion. In total 14 errors occurred, of which 8 were related to unidentified/ completely wrong substance (57.14%) and 6 were related to structural analogs and isomers (42,86%). I have added this information in percentage in the discussion. 

iii) In the Conclusion, please correct the typo "standard";

The typo has been corrected. 

iv) Why is not established a fixed protocol for these analyses?

As highlighted in the method, it was not the intention of the study to change the protocol within the DCS. They all consist of different analytical resources due to budgetary reasons. In sign of that, it was the intention to ameliorate the current way of working and improve this for the better. A sentence was added in 2b of 'materials & methods' to make sure this is clear for the reader. 

Reviewer 2 Report

Dear authors,

the manuscript “Challenges in drug surveillance: strengthening the analysis of 2 new psychoactive substances by harmonizing drug checking 3 services in proficiency testing” highlights the needing of an urgent, deeper study on the field of the emerging world of NPS, representing these substances a growing burden for global health and society. As widely explained in the paper, identification and monitoring of NPS could help to improve patient management, drug policies and market surveillance.

This study properly evaluates existing techniques via proficiency testing of European laboratories (DCS) on finding NPD, highlighting nodes of strength and weakness of current NPS research and surveillance strategies, besides proposing useful ideas and critical suggestions on possible solutions to improve present limitations.

 The manuscript is well written and clearly explained in every section.

Here are some minor suggestions that authors could consider in order to improve their manuscript.

1) Many studies are unveiling a growing abuse of different kind of substances with recreative purposes. They are mainly sold in on-line markets, usually as “legal drugs”, but it has been observed a growing trend in over-the-counter drugs abuse which should also be considered as a problem in detecting substances in intoxicated subjects, in some cases incurring in harmful collateral effects. To briefly outline this complex condition could add depth to this already exhaustive introduction.

e.g.:

Peacock, A., Bruno, R., Gisev, N., Degenhardt, L., Hall, W., Sedefov, R., White, J., Thomas, K.V., Farrell, M., Griffiths, P. 54405795600;35091057400;15769145500;7004040033;7402629359;8313658400;7405246388;57203810448;50961122100;57204078432; New psychoactive substances: challenges for drug surveillance, control, and public health responses (2019) The Lancet, 394

Graziano, S., Anzillotti, L., Mannocchi, G., Pichini, S., Busardò, F.P. 23995162800;36089049000;55701542100;7006708779;55931641600; Screening methods for rapid determination of new psychoactive substances (NPS) in conventional and non-conventional biological matrices (2019) Journal of Pharmaceutical and Biomedical Analysis, 163, pp. 170-179

Chiappini, S., Mosca, A., Miuli, A., Semeraro, F.M., Mancusi, G., Santovito, M.C., Di Carlo, F., Pettorruso, M., Guirguis, A., Corkery, J.M., Martinotti, G., Schifano, F., Di Giannantonio, M. 54392883800;57220441641;57210791249;57443080500;57443272200;57222627579;57191433783;36192150300;55496109900;7003858061;14060551000;7003711214;6603554568; Misuse of Anticholinergic Medications: A Systematic Review (2022) Biomedicines, 10 (2)

2) It could be useful, in Material and Methods section, to clearly state which and how many laboratories are included in the cited “Registered Laboratories”. In fact, it appears unclear if registered laboratories are only the ones included in TEDI network.

3)  Given the current huge number of NPS, examining a small number of them (20 in this study) could be a reduction of the problem, and it may lead to miss some analysis issue. Thus, this should be clearly stated as a limit of this study, affecting its generalization power.

Kind regards.

Author Response

1) Many studies are unveiling a growing abuse of different kind of substances with recreative purposes. They are mainly sold in on-line markets, usually as “legal drugs”, but it has been observed a growing trend in over-the-counter drugs abuse which should also be considered as a problem in detecting substances in intoxicated subjects, in some cases incurring in harmful collateral effects. To briefly outline this complex condition could add depth to this already exhaustive introduction.

The first article that was suggested, was already included in the article. The second one supports the statement that 'keeping up with NPS and the corresponding analysis is difficult' and was therefore added to the introduction. The comment of NPS being present in OTC medicine is valid, however the complexity of the NPS market is highlighted in different ways already in the introduction. Many examples are giving supporting the difficulties of keeping up with the market, as well as the presence of NPS in medicine such as xanax (alprazolam). In sign of that, the article was not added to the introduction. 

2) It could be useful, in Material and Methods section, to clearly state which and how many laboratories are included in the cited “Registered Laboratories”. In fact, it appears unclear if registered laboratories are only the ones included in TEDI network.

The term 'registered' was deleted and replaced, to not create confusion. The names of the 4 participating laboratories was also repeated. From the introduction it should be clear that Scanner harmonized 4 drug checking services from the TEDI network.

3)  Given the current huge number of NPS, examining a small number of them (20 in this study) could be a reduction of the problem, and it may lead to miss some analysis issue. Thus, this should be clearly stated as a limit of this study, affecting its generalization power.

A small paragraph was added to put emphasis on this in the discussion. It should also be noted that the article already points out that the study is limited:

  • Conclusion: limited proficiency testing
  • Participating services have access to adequate analytical resources to a certain level
  • conclusion: repetition of proficiency testing is necessary to further tailor the analysis